# The reality of embedded drug purchasing practices: Understanding the sociocultural and economic aspects of the use of medicines in Bangladesh

Md. Shahgahan Miah[1]*, Penchan Pradubmook Sherer[2], Nithima Sumpradit[3], Luechai Sringernyuang[2]

1 Department of Anthropology, Shahjalal University of Science and Technology, Sylhet, Bangladesh,
2 Faculty of Social Sciences and Humanities, Mahidol University, Salaya, Thailand, 3 Medicines Regulation Division, Food and Drug Administration, Ministry of Public Health, Yasothon, Thailand

* shahgahan-anp@sust.edu

**Data Availability Statement:** Primary data cannot be publicly available as data contains some

## Abstract

### Background

Purchasing drugs with or without prescription from retail drug shops is common practice in Bangladesh. However, what actually takes place between the drug seller and customer during the transaction is under-researched. This study explores the drug purchasing practices which underlie the socio-cultural and economic aspects of a Bangladeshi city.

### Methods

Adopting ethnographic methods, we conducted thirty in-depth interviews (IDIs) with customers, patients, and sales assistants, and ten key informant interviews (KIIs) with drug sellers, experienced sales assistants and pharmaceutical company representatives. Thirty hours were spent observing drug sellers' and buyers' conversations and interactions for medicine. A total of 40 heterogeneous participants were purposively selected from three drug stores. Transcribed data were coded, and analyzed thematically.

### Results

We found through thematic analysis that some individuals visited the drug store with fixed ideas about the name, brand, and dose of the drugs they wanted. Among the 30 IDIs participants, most individuals come without any preconceived ideas, describe their symptoms, and negotiate purchases with the expectation of quick remedies. Cultural practices of buying medicines in full or partial course of doses, with or without prescription, trust in sellers, and positive previous experiences of medications shape the drug purchasing behavior, regardless of any preconceived ideas concerning brand name, and dose. Few customers (n = 7) sought drugs by trade name, but most drug sellers often offered a generic substitute because selling non-brand drugs is more profitable. Notably, many of the clients (n = 13) bought drugs through installment payments and with loans.

sensitive issues and restrictions regarding participant consent about data share. The interested researcher can reasonably request Mr. Md Jitu Miah (Email: jitusust@gmail.com), Administrative Officer, Department of Anthropology, Shahjalal University of Science and Technology, Sylhet, Bangladesh. Interview guidelines are available in the Supporting Information files.

**Funding:** Yes partial for conducting the fieldwork but the funder have no role in study design, data collection and analysis, decision to publish or preparation of the manuscript.

**Competing interests:** The authors have declared that no competing interests exist.

## Conclusion

Community people choose and purchase the most necessary medicines in a self-medicated way from shortly trained drug sellers that can harm individuals' health and reduce the effectiveness of medication. In addition, the results of buying medicine through installments and loans suggest further research on the financial burden of consumers' purchasing behavior. Policymakers, regulators, and healthcare professionals might implicate the study findings to deliver practical information on the rational use of medicines to sellers and customers.

## Introduction

Global drug purchasing patterns and the way drugs are used in response to illness changed significantly with access to services, the availability of medicines, and aggressive competition among the drug firms [1]. The World Health Organization (WHO) reported that resilient healthcare systems are intrinsically linked to improved access to medication by ensuring rational selection, affordable prices, reliable supply systems, and sustainable financing [2]. One-third of the world's population lacks regular access to essential drugs; 50% of whom are in Asia and Africa [3]. In the South-East Asian Region, at least 65 million people are underprivileged in terms of access to essential medicines and vaccines [4]. Moreover, every year, US$ 42 billion is spent on the inappropriate use of drugs globally [5]. The severity of adverse reactions in low-and-middle-income countries (LMICs) was alarming, with 134 million adverse reactions occurring annually, resulting in 2.6 million deaths [6]. Therefore, WHO launched its third Global Patient Safety Challenge to reduce the level of severe, preventable harm related to medication by 50% over the next five years [5].

The pharmacy is the one place where drug purchasing and pharmaceutical procedures are discussed and negotiated by pharmacists and customers [7]. Modern medicines allow the pharmacist to prescribe drugs for primary care [8]. Therefore, the pharmacist can play a crucial role in delivering primary healthcare and the management of medication that contributes to improving health outcomes by providing clinical services, reducing healthcare costs and traveling and waiting times for a consultation [9–11]. In addition, the pharmacist can help to educate patients and guide healthcare staff on the use of medication, its efficacy, how long the drugs should be taken and the potential side effects [10,12]. Research revealed that individuals used to visit the retail drug shops directly to purchase drugs, and health supplements and seek advice during their most illness [13]. Dispensing and purchasing antibiotics without a doctor's prescription is often regarded as the most effective therapy for quick health recovery and financial constraints were reported for an incomplete dose of prescription drugs, even antibiotics [14,15]. Patients in Bangladesh also prefer to visit the pharmacy because of the availability of credit, flexible opening hours, and discounts [16,17]. In contrast, low priced and expensive brand drugs are dispensed according to the physical appearance of the customer [18]. A recent study observed that 88% of pharmacists recommend and put more effort into selling a generic substitute [19]. Previous studies have focused on the rationale of drug purchasing from the pharmacists' point of view but there is a lack of research from the customers' perspective that is crucial for ensuring the rational use of drugs.

Majority of patients in developing countries like Bangladesh, the nearest and only source of medicines is the retail drug store outside of the home [20]. 1,06,919 licensed drug shops have been identified while it was suspected there are an equal number selling drugs without a license [20]. The study found that 80% of people visit retail drug shops or poorly trained quack

doctors, a high proportion of whom (76%) self-medicate, and that 92% of drug sellers are 'C' grade pharmacists having only three months of training [20,21]. The Bangladesh Pharmacy Council and Bangladesh Chemist and Druggist Association grade pharmacists "A" for bachelor's degrees, "B" for diplomas, and "C" for three months of training. Notably, 95% of drug sellers are not concerned about the possible side effects of drugs that their patients/customers may suffer [16]. There is plenty of opportunity to prescribe, dispense, and take non-essential and unsafe drugs because of the lack of professional education and training, the desire for profit, and because they are easy prey for drug firms. The underlying factors of selling inappropriate antibiotics are related to a lack of knowledge, access to healthcare facilities and services, a lack of enforcement policies, and high dependencies on drug firms' information [22,23]. Drug purchasing practices might be influenced by sociocultural and economic aspects that contribute to unnecessary drug buying. Thus, this study set out to understand the sociocultural and economic aspects of drug purchasing in a drug store setting. It is hoped that our finding will aid policymakers in reinforcing the rational use of medicines in primary healthcare settings. In addition, data gathered in Bangladesh will contribute at the global level to efforts to achieve universal health coverage (UHC) as well as sustainable development goals (SDGs) 3.8 to provide "access to safe, effective, quality and affordable medicines and vaccines for all"

## Methods and materials

### Study time and settings

This ethnographic research was conducted between October 2020 and May 2021 in the *Shahjalal Uposohor* area in ward 22 (administrative zone) in the Sylhet Metropolitan area of Bangladesh. Sylhet city was identified as being densely populated (1045 of people living in per square kilometer) and having the poorest healthcare indicator in Bangladesh [24]. Within the study areas, high-income and low-income groups live in the suburbs and the '*Tero Ratan slum*' *respectively*. It has a population of 15,799 with a density of 2,074 and 32 healthcare centers with private retail drug shops, drug stores with a doctor's consulting room and clinics [25]. Following field study techniques of organizing complex research subjects and data, three topics were chosen as collection tools, namely 'places, people, and events, we selected three private retail drug stores to explore the actual drug purchasing practices using interviews and observation data [26]. The sociocultural and economic aspects include the expectations, negotiations, and decision-making processes regarding the sale and purchase of medicine that take place between drug sellers and customers in retail drug stores.

### Study population, sample and data collection techniques

In order to recruit study participants, the first author (SM) spent time at the drug store to observe and meet the customers who visited for the purpose of seeking primary healthcare and buying medication. From observation, we found that customers visited mostly at the morning (8 to 10 a.m.) and evening (5-7pm.). In the morning, clients visited for the purposes of diagnose the diabetes, measure blood pressure, and purchase medicine, while clients consult for primary healthcare, sought medicine, as the drug seller mostly available in the evening at the drug store. During the first week of fieldwork, the researcher used to wait at the front counter to observe because the sellers would not allow him to sit inside, possibly because there are sensitive issues around the selling and buying of drugs. However, the researchers' previous qualitative research experiences, student identity, and official letter from Mahidol University confirming that they were collecting data for research purposes, helped to build a good relationship with the drug seller and customers. A week later, the sellers voluntarily invited them to sit in and observe some initial conversations with customers. Participants were purposefully

selected from three different retail drug stores where they worked as private merchants, primary healthcare professionals, or consumers. Of the three drug outlets, one each of a seller with 'C' grade (having three months training) pharmacist, a rural medical practitioner (RMP) cum drug seller, and a trainee sales assistant. The customers who visited these retail drug shops did so either for their own health needs or for those of family members and relatives. Before recruiting participants for interview, we made a note of the customer's visits with the nature of their health problem, and whether they were asking for medicines with or without a prescription. The participants were invited for interview at a time convenient to them after observing the interaction which took place when they purchased their drugs. We recruited 40 participants following the principles of sampling parameters of settings, actors, events, and processes in line with the research questions and objectives [27]. We carried out a total of thirty in-depth interviews (IDIs) with patients (10), clients (15), trainee sales assistants (5), and ten key informant interviews (KIIs) with drug sellers (3), experienced sales assistants (2), medical representatives (3) and regular customers (2) (Table 1) in order to get a deeper understanding and contextualized data. In addition, it has been noted that carrying out observation helps in understanding the seller's and buyer's actual interaction behaviors when it comes to conducting interviews. The size of the sample was determined by following the principle of data saturation until reaching the point of repetition and nothing (data, theme, and dimension) new being found [28]. Saturation level was identified by analyzing the data as an iterative process during the fieldwork.

**Table 1. Socio-demographic background of the participants.**

| IDI (n = 30) | | KII (n = 10) | |
|---|---|---|---|
| **Characteristics** | | | |
| **Residents** | | | |
| *Suburb* | 15 | *Suburb* | 8 |
| *Slums* | 15 | *Slums* | 2 |
| *Participants* | | | |
| *Client* | 15 | *Drug seller* | 3 |
| *Patient* | 10 | *Medical representative* | 3 |
| *Trainee sales assistant* | 5 | *Experienced sales assistant* | 2 |
| - | - | *Regular customers* | 2 |
| | **IDI** | **KII** | |
| **Age in years (Mean ± SD)** | **40±10** | **45±9** | |
| **Sex** | | | |
| *Male* | 25 | *Male* | 10 |
| *Female* | 5 | - | |
| **Education** | | | |
| *0–5* | 7 | *6–10* | 2 |
| *6–10* | 5 | *11–12* | 3 |
| *11–12* | 8 | *13+* | 5 |
| *13+* | 10 | - | - |
| **Profession** | | | |
| *Day labor* | 8 | *Provider cum seller* | 3 |
| *Rickshaw puller* | 5 | *Medical representative* | 3 |
| *NGO worker* | 4 | *Experienced sales assistant* | 2 |
| *Govt. service holder* | 3 | *Regular clients* | 2 |
| *Teacher* | 5 | - | - |
| *Homemaker* | 5 | - | - |

## Data collection procedures

We conducted the interviews based on the participants' voluntary participation, availability and choice of a suitable place. Some interviews were conducted in a private area in the drug store, and some took place elsewhere such as at the customer's home or in a park. We explained about our research, the reason for carrying it out and how the findings would be used when recruiting participants. Semi-structured interview guidelines were developed in order to keep the interviews on track which covered a range of issues such as the drugs requested, consultations, dispensing patterns, cost and negotiations over buying drugs. All interviews were recorded using a digital recorder that ran for 45–60 minutes, although two participants did not agree to recording. Observation data captured the conversations and interactions of local cultural practices of medicine buying full or partial course of drug dose with or without prescription which might not be revealed in the interviews. These observational data help to compare, justify and triangulate with the data collected in interviews. All the interviews were conducted in Bengali because it was the researcher's and study participants' first language.

## Data analysis

Data was transcribed verbatim and carrying out member checking for participant validation to share the data with participants [29]. Then, transcriptions were translated into English to ensure consistency of meaning. All four authors were involved in generating open codes manually following research objectives independently. Then an inductive process was followed to analyze the data thematically from the emerging patterns of drug purchasing themes and sub-themes and methodologically triangulated in order to understand the uniformities of the data [30,31]. Thus, drug transactions between the customer and seller, and the medicines used are described and analyzed as a social process.

## Ethical issues

This study has ethical approval from the Mahidol University Central Institutional Review Board (MU-CIRB), Thailand, protocol number MU-CIRB 2020/237.3108. Ethical issues ensured that those taking part are not put at any risk of trouble or finding themselves in a harmful situation from participating in this study. Both verbal and written consent was obtained before the interviews were initiated. A consent form explaining the study's purpose, the participant's rights and guarantee of confidentiality was briefly but carefully read out prior to starting each interview. A pseudonym was used to protect the identity of the participants.

# Results

We conducted 40 interviews with the study participants (Table 1). The mean age of participants was 40 (SD ±10) for IDIs and 45 (SD ±9) for the KIIs participants. The majority of participants had post-secondary education but participants from slums had only received primary education. All participants, except two, were married. IDIs participants were from a wide range of occupations: eight were labors, five rickshaw pullers, five worked in non-government organizations (NGO), three in the public sector, and five were teachers, whereas all the female participants were homemakers. KIIs participants were directly involved with selling and prescribing drugs and worked in jobs such as healthcare service providers, drug sellers, medical representatives and experienced sales assistants, whereas two were long standing customers who regularly purchased and took medicines.

**Table 2. Demographics of retail drug stores: Insights from drug shop staff and customers.**

| Description | Drug store 1 | Drug store 2 | Drug store 3 |
|---|---|---|---|
| Location | Higher income residential area | Slum area | Commercial bazaar area |
| Sellers | Rural medical practitioner | Sales assistant and occasionally seated health assistant | 'C' grade pharmacist (3 months training) and sales assistant |
| Clients | Higher economic, educated and service holder | People living with poverty are rickshaw puller, day labor | Higher income, educated, service holder, and people living with poverty |
| Commonly reported health problem | Diabetic, blood pressure, heart diseases | Gastric, Pain, Diarrhea, common cold and fever | Diabetic, blood pressure, diarrhea, pain, accident patient, cold, fever |
| Products Sold | Brand drugs, imported medicines | Non-brand, low-safety, allopathic, herbal drugs | Brand, imported, non-brand, low safety, herbal |

## Context

We found various types of retail drug shops in the study areas considering the location, sellers training and the client flow. We purposively selected three drug stores located in three different settings (Table 2). The first was located on a commercial road side and was operated by a seller with limited training ('C' grade seller having three months training). The second was in an apartment building residential area and operated by a rural medical practitioner cum drug seller. The third one was located next to the slum areas and was operated by a sales assistant and a health assistant (working at the community clinic), who often visits in the evening. All three retail drug shops were brick-walled and tin-shed houses, and the front space was opened for the patients. Only one drug shop had a refrigerator for storing medicine and privacy spaces for the clients (Fig 1). Those who dispense drugs are described as a 'drug seller' because of their poor professional training. The regular opening hours for dispensing drugs at all three

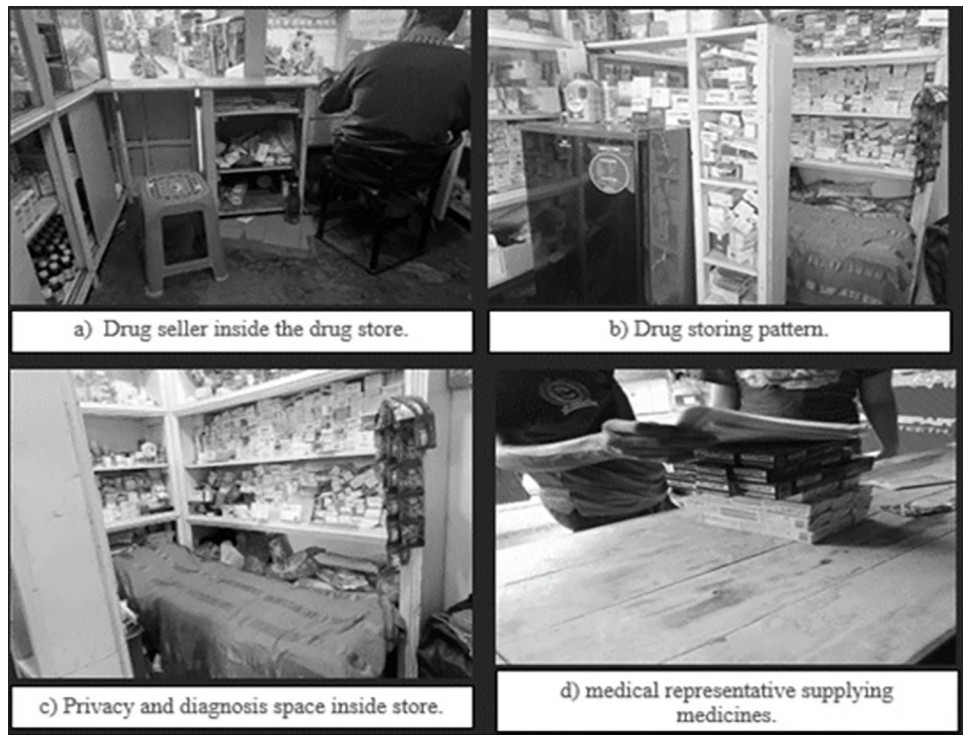

**Fig 1. Drug shop in Bangladesh.**

drug stores were between 8 a.m. and 10 a.m. In addition, the sellers frequently open early and close around 11p.m. for their clients' emergency requests. However, the drug seller visited customers' home on an emergency call. Customers mostly visited the drug store without a prescription, had previous experience of self-medication, and consulted the drug seller in order to select medicines. In addition, drug stores in Bangladesh provide other services such as photocopying, mobile phone top ups, and mobile banking like other superstores, as well as dispensing and selling drugs. Poor regulation and monitoring, a shortage of health workers, and a limited budget for healthcare all influenced the way medicines are bought for self-medication. The way in which customers purchase medicine is influenced by cultural practices about medicine usages, positive previous experiences, trust in the seller, and the social network of the community.

Several drug sellers mentioned that those living in apartment building areas mostly bought prescription and trade name drugs. However, people from the slum areas sought healthcare and purchased medicines on the recommendation of the sellers. The customers whovisited these drug shops did so either for their own health needs or for those of family membersand relatives, and they came with or without any preconceived ideas about the name, brand anddose of the medicines they wanted.

In the following sections, we categorized three key themes of drug purchasing interactions (Fig 2): the purchasing practices, sociocultural and economic reasons of medicine tradeoff discussed below.

## Drug purchasing practices

**People visit with preconceived ideas about the name, brand and dose of drugs.** Our data revealed that apartment building residents visited the drug store with certain preconceived ideas about the trade name, brand, and dose of the drugs they wanted. Drug purchasing

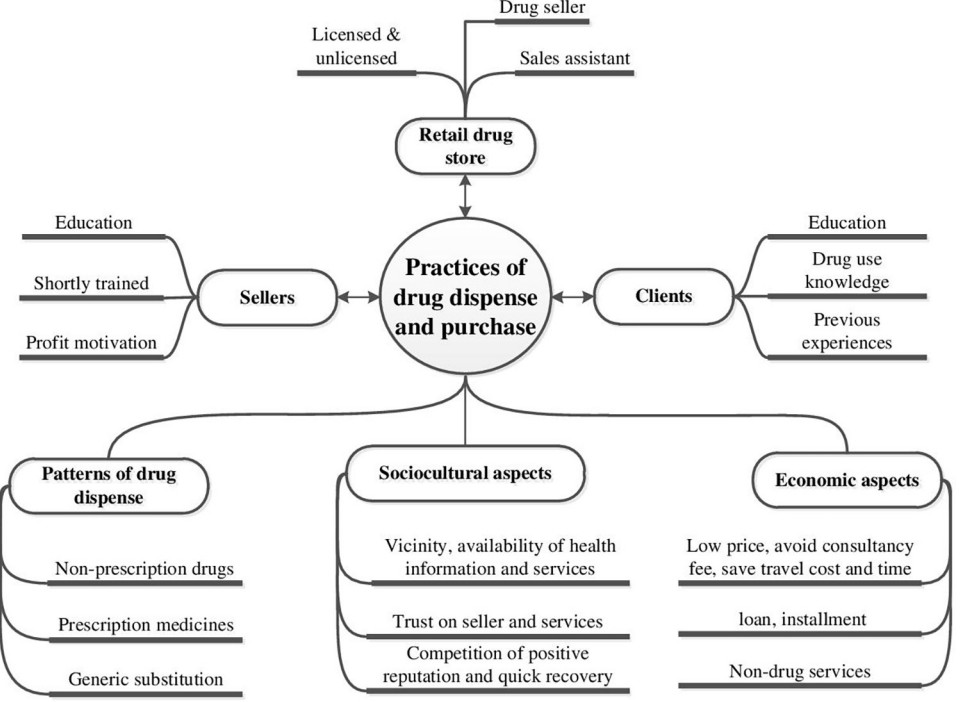

**Fig 2. Emerging codes and aspects.**

behavior was influenced by previous satisfactory experiences of using these drugs. Study participants reported that the drug purchasing behavior of individuals, who seek primary healthcare with preconceived ideas, includes asking for particular drugs by trade name and bring their new and their old prescription to show to the seller. In particular, drugs previously prescribed are picked up together with a full course of medicines. One participant stated in IDI:

"I have been taking these medicines for three years. I feel good and have no complaint regarding this. So, I mostly bought one month's drugs at the same time".

In contrast, participants reported that the customer purchases only half and part of medicines recommended in case a new prescription is given, calculating how much to buy and negotiating the cost of the drugs. One customer shared the issue in IDI:

"Doctor asked me to visit again after a week. If I do not feel better within a week, the doctor might change the list of medicines during the next visit. That will be a huge loss as the medicines already bought can't be returned".

**People who visit the drug store without any ideas about medicines they want.** Seeking healthcare and drug purchasing behavior among people from the slum is influenced by the customer's individual wishes, previous experiences, and by the fact that they had similar symptoms to friends or family members. Participants reported that the participants, mostly low-income and illiterate customers in the drug shop 2 only, asked the sellers about which drugs to take by describing their symptoms, and setting the price of the drug for the ailment. Observation revealed:

"Give me a five takas' gastric drugs"

Data from in-depth interviews revealed that participants knew the use of particular drugs and they regularly used those. Participants often purchase a drug by trade name, even antibiotics. One customer elaborated in IDI:

"I have always asked for cef3 (azithromycin) for my sons' cough and fever for the last four years. And, they used to get well with this antibiotic. I do not need to ask the drug seller about medicines."

**Drug purchased with a loan and by installment payments.** Study participants considered the retail drug store (drug shop 2) as the poor people's hospital, a place they could easily access and where they could seek advice and purchase medicine without having to pay consultancy fees and travel costs. Furthermore, participants can purchase drugs here with loan. At the end of the month, the customers from the slums pay their medicines bill like other household bills. This arrangement increased trust in the seller and the use of drug. One participant stated his experiences in IDI:

"I just asked for the drug by name and often by ailment, whatever I need. *Tapan* (sales assistant) writes the amount in the loan book. The day I get my salary I paid the drug bill in cash".

In addition, the customer also seeks to buy a package of expensive drugs by installment payments as well as by buying recommended drugs and those for a particular ailment. Most of the

low-income participants purchase daily packages of drugs on the way home from the work. Participants reported that there was a mutual understanding about the installment of medicine and the payment between the sellers and clients as clients often do not buy a complete package of medicine at once but rather buy on a daily basis installment with hand cash payment. The following conversation was recorded during observation:

> "I have been suffering from a fever and have pain all over my body" said the customer. The seller calculates a total of 250 takas for a five-day course of drugs at 50 takas a day (around $3 @ 85 takas per USD) using a calculator and asks customers for a decision. The customer replied he would take one-day's supply and will buy some each day on the way home after finishing work" (Drug store, *Teroraton* areas)

In contrast, paying by installment might encourage the irrational use of drugs as the customer stops taking them when he feels better. One provider cum drug seller explained the customers' behavior:

> "Some customers cannot afford to buy a full course of drugs. Often they stopped buying if they feel better. If I (seller) asked, the customer **often** replied that they no longer have a problem so they don't need to take any more medicine."

**Drug sellers put more effort into the substitution of generic drugs to maximize profit.** The efforts put into substituting generic drugs emerged as an important aspect in the process of purchasing drugs. Participants reported that the seller put more effort into substituting generic medicines when the medicines the customers asked for by trade name were not available. In particular, medicines are commonly requested and purchased according to the trade name. Therefore, the seller offers alternative pharmaceutical products in order to maximize profit. Even if the customer asked for a specific brand by trade name, the seller will try to convince the customer to be guided by his expertise on drugs. One observation reported the following conversation during the purchasing of drugs:

> Customers asked for *flucox* (one brand name of antibiotic), the seller dispensed *G-flucloxacillin (another brand name)*. When the customer refused to buy the substituted medicines, the seller said that "the only difference is the name of the drug, they are the same as two brothers with the same mother."

Data also identified that if customers want to go to another drug store if they do not agree to the alternative offered, the seller asked them to wait and got the desired brand from another drug shop.

**Dispensed non-brand and substandard drugs.** Some of the drug sellers reported that they dispensed and sold some unregistered (not registered with the drug administration) local drugs in the drug store in the slum. The drug seller or sales assistant dispensed non-brand and substandard drugs in order to maximize profit and did not care about the quality of these medicines. For example, drug stores in slum and commercial areas recommended and dispensed these substandard, non-brand drugs to people who visited without any fixed ideas about which drugs they wanted. Customers did not know about the drug's quality, brand name, and expiry date because they trusted the seller. One drug seller stated in KII:

> "The medical representative offers more profit and a higher commission on the non-brand products. As we do not take a consultation fee and have no other source of income, except

the profit from drug selling, we tried to sell these drugs to the customer who asked us for medicines, especially the rickshaw pullers and daily laborers".

**Expectations and negotiations in purchasing drugs.** Customers' expectations of the medicines, how much they cost and whether it is possible to negotiate the price, influenced drug purchasing behavior. The customers expect to get well quickly after taking the medicines. One sales assistant explained:

"Customers just want to get well immediately after taking the medicine."

Besides, the negotiation process between the seller and customers on the cost and dose of the drugs dispensed has an influence on the choice of drugs. One observation found:

"Customer: asked the seller for medicines for sneezing and a fever. Seller: dispensed a three-day course of antibiotics. But the customer requested a two-day course because of a lack of money. Then, the seller agrees with customer and suggests that he must complete the full course. The customer smiles and replies 'Of course I will."

## Drug transaction practices: The pharmacy as a comprehensive primary health care service

**Sociocultural aspects.** It is the availability of healthcare services and information at drug store that attracts customers. This includes the prescribing and selling of medicines, measuring blood pressure, testing for diabetes, checking for a fever, providing health information sources and referral to a hospital/clinic, visiting the hospital with customers, booking further appointments at physician's consulting room, home delivery services, and acting as a circumcision center. Participants reported that customers, especially older individuals, can avoid visiting the diagnostic center because of the availability of diabetes diagnosis facilities at their nearest drug shop. These facilities provide rapid testing of fasting blood samples and reduce the need for carers' visits and the cost and time taken to travel to the center. One participant stated:

"For a fasting blood test, my father has to wait for hours at the diagnostic center in a long queue, and he is used to having his breakfast early in the morning. Now, he can just give the sample to the nearest pharmacy, get the result, and can have his breakfast at home".

Our participants also reported that many older people and pregnant women who are prescribed medication by the physician that has to be administered by injection find this challenging and it is risky for them to go to the hospital. The drug seller is their only option for accessing this treatment. One female customer stated:

"I was about thirty-five weeks pregnant and my blood test identified a severe iron deficiency. My gynecologist prescribed seven iron injections. My spouse was worried to hear that I would have to travel to the hospital for the injections! Then the doctor assured him that any drug seller could do this. We got our neighborhood pharmacy doctor to do it".

**Trust in the seller and the services they provide: The pharmacist as a healthcare consultant.** Trust in the drug seller and their services emerged as an important reason for purchasing drugs from the drug store. We found that the customer often visited the drug store with no preconceived ideas and asked for medicines by describing their ailment(s). In particular,

participants described the drug store as the peoples' mini hospital and/or clinic, *Osuderdokan* (drug store), *daktarerghor* (doctor house), a referrals center for selecting a doctor and hospital with a referral from the drug seller. During observation, the following conversation was reported between a customer and seller:

> "I slipped over on the road two weeks ago and hurt my right elbow. What should I do?"
> **The customer** presents the symptoms while the **drug seller** replied, "You can go for an x-ray. It will cost 200 takas and then go to the doctor"

Furthermore, the drug seller is thought of as a trustworthy mentor for selecting the best possible physicians, clinic, hospital, diagnostic center and also visited the healthcare center with the customers in cases of emergency. A participant explained:

> "My father felt severe chest pain at midnight, and I was worried about where to go? Then my younger brother asked me to call the pharmacy *dakter* (doctor). Farhad *dakter* visited us with his stethoscope and blood pressure instrument. He did not stop at just referring us to hospital but also went there with us, consulted the doctor, and arranged a hospital bed."

**Competition to achieve a reputation for successful treatment.** Participants reported that trust in sellers and their services were linked to having a good relationship and good experiences of treatment. In addition, good experiences of treatment in a drug store setting are influenced by the time it takes to get better. Participants reported that not only is the seller motivated to cure ailments quickly in order to create goodwill but the patient or parents also want to get better immediately, even if it means taking high doses of antibiotics in most cases. Participants reported that this competition might increase the overuse of antibiotics. One drug seller explained:

> "If I do not give antibiotics, the patient will not get better quickly because we cannot diagnose diseases. Antibiotics work for any kind of illness although it may harm your health".

One medical representative shared the fact that drug sellers competed among themselves to build a good reputation in the community that would help to maintain their drug business.

> "There is competition to create a good reputation for curing disease as quickly as possible. They do not think about the side effects of the drugs, and even the buyer wants to get well quickly. There is competition to provide antibiotics in every case."

## Discussion

This study investigated the selling and purchasing practices between drug sellers and customers in retail drug store settings in Bangladesh. Drug stores are considered comprehensive primary health care service points at the community level; because they offer a range of services—such as easing access to the purchase of physician-prescribed medicine, drug seller based treatment, measuring blood pressure, diagnosing diabetes, providing health information, making referrals to clinics or hospitals, booking physician appointments, and offering home delivery of medicine and treatment services to the community. We found evidence that there are plenty of unlicensed retail drug stores prescribing and selling drugs with short-trained drug sellers and sales assistants [8,20,21]. Residential status, together with cultural practices about medicines usages and social networks, determines the drug purchasing practices of the people living

in slums and apartment buildings. This study showed that people negotiate with a fixed idea of the trade name, brand, and dose of the drugs they want, ask for drugs according to cost and ailment, bring old prescriptions and samples to show the seller, and discuss what the medicines may treat before deciding what to buy. Patients often bought a partial course of medicine with a new prescription to check whether the medicine improved their health or not. Most sellers and customers preferred antibiotics for quick remedies, and incomplete doses were widespread [14,15]. The drug sellers and sales assistants recommended and dispensed non-brand and substandard drugs to the clients from the slums. In addition, the drug seller tries to substitute medicines, whether the individual comes with preconceived ideas or has no ideas about the drugs they want. Drug sellers even collect the desired drugs from adjacent drug stores if they do not have the drugs demanded. Without doubt, this was to maximize their profit [18–20]. The study revealed that both sellers and clients lack knowledge about the medicine's use, side effects, and quality [22,23].

The sociocultural aspects of buying drugs in the drug store include the availability of information and services at the doorstep that save the time and cost of travelling and consultancy fees, as reported by other studies [9–11,22]. This study explores similar findings to other studies carried out in Bangladesh that the nearest and only source of medicine purchase is the pharmacy [20]. Our findings also reported that, without doubt, the reason for seeking medical care and the medicine purchasing practices was the proximity of services which attract people in the community, not only to the drug store but also to the drug seller who will go to the patient's home. For instance, the drug seller often made home calls to the elderly and pregnant women to carry out check-ups and give injections and saline infusions. In addition, drug sellers played the role of health educator having recommended physicians, hospitals, and clinics to those worried about their health after listening to their symptoms that shapes the clients' medicine purchasing preferences. However, the sellers did not provide information on the efficacy, duration, and potential side effects as suggested by previous studies carried out in developed countries [10,12]. Familiarity with the pharmacist, good experience with treatment and satisfaction with their services create trust in the provider and their services, which, in turn, influences the seeking of care and the purchasing of drugs. For example, curing an ailment immediately creates a good reputation for having a special ability to treat illnesses. In particular, drug sellers offer all kind of treatment to over-the-counter patients with a high dose of antibiotics to gain community favor and sell more medications. In addition, patients and/or parents often avoid getting a diagnosis of their disease and use a course of well-tried antibiotics as 'powerful' medicines which can treat a wide range of ailments. These findings might assist policymakers in formulating policy to monitor and regulate the over-the-counter drug according to the national drug policy and multilayer interventions to ensure the safe use of good quality medicines.

The study also revealed that patient behavior towards visiting the medicine shop was due to the availability of credit, flexible opening hours, and discounts on the price [16,17]. Hence, the practice of purchasing drugs with a loan and through installment payments is a new aspect of drug buying which has increased access to medicines, especially for the underprivileged. Notably, the drug sellers feel more comfortable recommending unregistered drugs to lower socioeconomic classes—for example rickshaw pullers and day laborers. However, there is some risk of misuse of drugs because providers and sellers are poorly trained and there is a lack of awareness among customers. For example, a customer may discontinue taking the antibiotics when they feel better, which may lead to antibiotic resistance and the misuse of drugs [16]. In addition, drug sellers are not concerned about customers when it comes to the possible side effects of drugs. These findings about the beliefs, culture and experiences of practices relating to the use of medicines will increase access to medicines with patients being aware of the proper use

of drugs and their possible side effects, as well as improve the quality and safety of drugs implementing multilayer interventions by the policymakers. In the context of the shortage of health workforces, both patients and pharmacists will be in a position to adopt practices that improve primary healthcare in Bangladesh.

### The limitations of the study

This study focused on the sociocultural and economic aspects of drug purchasing practices in the urban setting, but seeking medical care and medicine purchasing practices might vary widely in the rural context of Bangladesh. Furthermore, sensitivity about the whole issue of recommending, dispensing and selling drugs meant that it was difficult for the sellers to allow us, as outsiders, into their professional space, and observe the services provided at the drug store. The risk was that this might affect the customer's trust. However, the reality of health and drug related practices are context bound, and so these findings can be applied to other places of drug purchasing.

### Conclusion

This study depicted that drug purchasing practices are influenced and shaped by the close proximity of drug stores that eases medicine accessibility, sellers' training, customer's seeking healthcare and social network. Customer's socio-cultural practices of purchasing medicine depends on trust in the sellers, previous medication experiences, economic aspects such as buying drugs with a loan or by installment payments, and avoiding consultancy fees. Individuals buy medicines based on their needs, either with a prescription or for self-medication, by purchasing partial course of medication and do not necessarily complete the recommended course of medicine. The drug seller visited clients in case of emergency call and often puts more effort into selling generic drug substitutes. These actions might be considered as the unlawful and unnecessary selling of drugs from the legal and pharmaceutical points of view but customers consider their drug purchasing and use of medicine is rational because of lack of access to physicians, healthcare facility center, and awareness. Both the customers and sellers are violating the rules of drug selling, buying and using prescription only medicine without prescription practices. Government can train and educate sellers and clients to improve medicine dispense, sells and purchase practices. Monitoring and regulation alone is not enough to ensure the supply of safe, good quality and rational use of medicines. Thus, the local context of sellers and customers' drug practices should be considered in order to ensure the rational use of medicines and a sustainable healthcare system and services as drug selling and purchasing practices varied within the urban settings of slum and suburb areas.

### Supporting information

**S1 File. Research tools in-depth interview and key informant interview guidelines, observation checklist.**
(DOCX)

### Acknowledgments

We would like acknowledges our study participants for their volunteer participations.

### Author Contributions

**Conceptualization:** Md. Shahgahan Miah, Luechai Sringernyuang.

**Data curation:** Md. Shahgahan Miah.

**Formal analysis:** Md. Shahgahan Miah, Penchan Pradubmook Sherer, Nithima Sumpradit, Luechai Sringernyuang.

**Investigation:** Md. Shahgahan Miah.

**Methodology:** Md. Shahgahan Miah, Penchan Pradubmook Sherer, Nithima Sumpradit, Luechai Sringernyuang.

**Supervision:** Md. Shahgahan Miah, Luechai Sringernyuang.

**Validation:** Md. Shahgahan Miah, Penchan Pradubmook Sherer, Nithima Sumpradit, Luechai Sringernyuang.

**Writing – original draft:** Md. Shahgahan Miah, Luechai Sringernyuang.

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
