## [Decision Letter · Decision Letter 0]

10 Jul 2022

PONE-D-22-12723The Reality of Embedded Drug Purchasing Practices: Understanding the Sociocultural and Economic Aspects of the Use of Medicines in BangladeshPLOS ONE

Dear Dr. Miah,

Thank you for submitting your manuscript to PLOS ONE. After careful consideration, we feel that it has merit but does not fully meet PLOS ONE’s publication criteria as it currently stands. Therefore, we invite you to submit a revised version of the manuscript that addresses the points raised during the review process.Please address each comment made by the reviewer point by point

We look forward to receiving your revised manuscript.

Kind regards,

Mary Hamer Hodges, MBBS MRCP DSc

Academic Editor

PLOS ONE

Journal Requirements:

2. During our internal checks, the in-house editorial staff noted that you conducted research or obtained samples in another country. Please check the relevant national regulations and laws applying to foreign researchers and state whether you obtained the required permits and approvals. Please address this in your ethics statement in both the manuscript and submission information. In addition, please ensure that you have suitably acknowledged the contributions of any local collaborators involved in this work in your authorship list and/or Acknowledgements. Authorship criteria is based on the International Committee of Medical Journal Editors (ICMJE) Uniform Requirements for Manuscripts Submitted to Biomedical Journals - for further information please see here: https://journals.plos.org/plosone/s/authorship.

“Yes partial for conducting the fieldwork but the funder have no role in study design, data collection and analysis, decision to publish or preparation of the manuscript”

“None”

7. Please include a copy of Table 1 which you refer to in your text on page 8.

Additional Editor Comments:

You study is of interest and importance especially for Low Income Countries. However, as pointed out by the reviewer there are major revisions required in the presentation and interpretation of results. I recommend a thorough revision dealing which each comment before considering re submission to PLOS ONE.

Reviewers' comments:

Reviewer's Responses to Questions

**Comments to the Author**

1. Is the manuscript technically sound, and do the data support the conclusions?

Reviewer #1: Partly

2. Has the statistical analysis been performed appropriately and rigorously? 

Reviewer #1: N/A

3. Have the authors made all data underlying the findings in their manuscript fully available?

Reviewer #1: No

4. Is the manuscript presented in an intelligible fashion and written in standard English?

Reviewer #1: Yes

5. Review Comments to the Author

Reviewer #1: Comments

This article explored an interesting and important aspects of drug purchasing practices in Bangladesh. However, one of the important flaws of this paper it tries to emphasis the vague quantitative findings through qualitative method without taking into account any quantitative factors. I would suggest the author to focus on the findings on the reasons why this practices rather than emphasizing on the 30 background factors e.g., Income, education, cultural beliefs, confidence in sellers, among others to make a conclusion.

Abstract

1. The author mentioned that “This study explores patterns of drug purchasing which underlie the socio-cultural and economic aspects of a Bangladeshi city”. How is it feasible to explore pattern through a qualitative study?

2. The results seemed to be vague quantitative findings without any specific numbers.

Background

1. Line 46: “….aggressive competition among the drug firms” this should be supported by evidence.

2. Line 52: “Moreover, every year, US$ 42 billion is spent on the inappropriate use of drugs” Where?

3. Line 73: “For most patients in developing countries like Bangladesh…. what proportion? Any information?

4. Line 88: ““access to safe, effective, quality and affordable medicines and vaccines for all”” I do not understand how the author has linked "access" with the context of current manuscript. Is this author meant the availability of quality drugs? If yes, is availability of quality drugs in pharmacy is similar to access? My understanding is that ‘access’ is determined by various socioeconomic factors and different from availability.

5. Line 97: is not clear. What the author meant by density? Per facility?

6. Line 99: “Following field study techniques of organizing complex research subjects and data, three topics were chosen as collection tools, namely 'places, people, and events.” Should be supported by evidence of where this technique has been used?

7. Line 186-187: “The opening hours of these drug stores were between 8.00 am and 10.am, however, they visited customers at home in an emergency.” This line is not clear. What the author wanted to mean here?

8. Line 181: the author mentioned that “three different drug stores were selected from three different settings?” It is not clear how the author determined different settings in terms of what? and why the author have chosen three different settings?

9. Line 195: author mentioned that “Customers’ drug purchasing behavior was influenced by demographic factors such as income, education, cultural beliefs and social network of the community people.” it is predictable that these socioeconomic factors influence customers’ drug purchasing behavior. However, what is new here? It would be useful to identify to what extent which factors affect the behaviour of the consumers? Otherwise, this statement remains vague, and dose not add anything to the findings. I assume this may not be possible to quantify this finding from only 30 interviews. I would recommend author not to focus on only the factors those affect the consumer drug purchasing behavior rather identify the key reasons for the identified behavior.

10. Line 196: “Those living in wealthier areas mostly bought prescription and trade name drugs” I do not see any calculation of wealth quintile. How the author concluded this is not clear. A clear representation of the calculation of wealth quintile should be presented. Or on what criterion the author divided the respondents into different socioeconomic group should be mentioned clearly.

11. Line 207-208: please see comment 09.

12. Line 225: “customer’s individual choices” I am not sure how the author determined the choices of consumer? Did the author wanted to mean their previous behaviour of drug purchasing?

13. Line 246: “In addition, the customer also seeks to buy a package of expensive drugs by instalment payments as well as by buying recommended drugs and those for a particular ailment.” This finding is interesting. However, the author ha not mentioned what is the understanding between the consumer and the drug seller regarding EMI? How the expensive drugs will be sold to consumers in EMI without any guarantee of repayment by the consumers?

14. Line 252: “around $3 @ 85 takas per $” The author should mention which dollar they rereferred to.

15. Line 278: “Findings revealed that some unregistered local drugs were dispensed and sold in the drug store.” I think it is difficult to conclude this finding from information from only three drug stores.

16. Line 406: “patient preference” the preference is a technical term, which should be assessed as revealed or stated. I would rather suggest the author to use behavior instead of preference.

6. PLOS authors have the option to publish the peer review history of their article (what does this mean?). If published, this will include your full peer review and any attached files.

Reviewer #1: No

---

## [Author Response · Author response to Decision Letter 0]

11 Sep 2022

Dear Academic Editor/Mary Hamer Hodges, 

PLOS ONE journal

We sincerely thank you for the constructive comments and the opportunity to revise our manuscript ID: PONE-D-22-12723R1 entitled “The Reality of Embedded Drug Purchasing Practices: Understanding the Sociocultural and Economic Aspects of the Use of Medicines in Bangladesh’” for publication consideration in the journal of PLOS ONE. We made the revisions carefully and incorporate them into our revised manuscript in accordance with the comments. Our great pleasure is that the reviewer (s) have recommended our manuscript for publication with major revisions. Details of changes are listed as follows: 

Journal additional requirements.

Response: Checked accordingly. Thank you so much. 

2. During our internal checks, the in-house editorial staff noted that you conducted research or obtained samples in another country. Please check the relevant national regulations and laws applying to foreign researchers and state whether you obtained the required permits and approvals. Please address this in your ethics statement in both the manuscript and submission information. In addition, please ensure that you have suitably acknowledged the contributions of any local collaborators involved in this work in your authorship list and/or Acknowledgements. Authorship criteria is based on the International Committee of Medical Journal Editors (ICMJE) Uniform Requirements for Manuscripts Submitted to Biomedical Journals - for further information please see here: https://journals.plos.org/plosone/s/authorship.

Response: Thank you so much for your note. For conducting this research, the researcher got a request letter for permission to collect research data in Bangladesh addressing the head of the health sector at the district level for conducting the research (see the request letter for permission to collect research data as attachment) from Dean, Faculty of Graduate Studies, Mahidol University, Thailand. The district's head of the health sector approved the request to collect research data as the study had already received the certificate of ethical approval (Protocol No. MU-CIRB 2020/237.3108) from the Mahidol University-Central Institutional Review Board.

“Yes partial for conducting the fieldwork but the funder has no role in study design, data collection and analysis, decision to publish or preparation of the manuscript”

Response: Thank you so much for the queries. Yes, included the statement in the cover letter:” The funders had no role in study design, data collection and analysis, decision to publish, or preparation of manuscript”.

“None”

Response: Yes, we included the state: "The authors have declared that no competing interests exist." in the cover letter.

Response: Yes, there are ethical restrictions on sharing the data set according to the Mahidol University Central Institute of Review Board (MU-CIRB) as data contain potentially sensitive information. We revised this in our cover letter. 

Response: Thank you so much for the suggestion. We revised according.

7. Please include a copy of Table 1 which you refer to in your text on page 8.

 Response: We included the Table 1.

Additional Editor Comments:

Reviewers' comments:

Reviewer's Responses to Questions

Comments to the Author

1. Is the manuscript technically sound, and do the data support the conclusions?

Reviewer #1: Partly

2. Has the statistical analysis been performed appropriately and rigorously?

Reviewer #1: N/A

3. Have the authors made all data underlying the findings in their manuscript fully available?

Reviewer #1: No

4. Is the manuscript presented in an intelligible fashion and written in standard English?

Reviewer #1: Yes

5. Review Comments to the Author

6. PLOS authors have the option to publish the peer review history of their article (what does this mean?). If published, this will include your full peer review and any attached files.

Do you want your identity to be public for this peer review? For information about this choice, including consent withdrawal, please see our Privacy Policy.

Reviewer #1: No

 

Response to reviewer comments

Reviewer#1: Comments

This article explored an interesting and important aspects of drug purchasing practices in Bangladesh. However, one of the important flaws of this paper it tries to emphasis the vague quantitative findings through qualitative method without taking into account any quantitative factors. I would suggest the author to focus on the findings on the reasons why this practices rather than emphasizing on the 30 background factors e.g., Income, education, cultural beliefs, confidence in sellers, among others to make a conclusion.

Response: Thank you for your comment. In the revised version, we deleted the quantitative findings. 

Page no.: 2, line no. 30-31.

Abstract

1. The author mentioned that “This study explores patterns of drug purchasing which underlie the socio-cultural and economic aspects of a Bangladeshi city”. How is it feasible to explore pattern through a qualitative study?

Response: Thank you so much for the comment and suggestions. We revised the abstract as “This study explores the drug purchasing practices in place of patterns of drug purchasing. 

Page no.: 1, Line no. 16.

2. The results seemed to be vague quantitative findings without any specific numbers.

Response: Thank you so much for your comments. We revised and deleted the quantitative findings. 

Background

1. Line 46: “….aggressive competition among the drug firms” this should be supported by evidence.

Response: Thank you so much for the comment. We have checked and included the reference as no. 1, namely Seeberg (2012). Apologies for the mistake. Noted that this inclusion of the new reference brought changes to the numbering of the entire manuscript references. Thanks for understanding.

Page no.: 3, Line no.: 46.

2. Line 52: “Moreover, every year, US$ 42 billion is spent on the inappropriate use of drugs” Where?

Response: We have checked the reference and corrected it as globally. 

Page no. 3, Line no. 52.

3. Line 73: “For most patients in developing countries like Bangladesh…. what proportion? Any information?

Response: Thank you for the comment. We checked the reference and found that the article reported the proportion as the majority of the patient. We included ‘majority of’ and ‘outside of the home.’ 

Page no. 4, Line no. 74-75.

4. Line 88: ““access to safe, effective, quality and affordable medicines and vaccines for all”” I do not understand how the author has linked "access" with the context of current manuscript. Is this author meant the availability of quality drugs? If yes, is availability of quality drugs in pharmacy is similar to access? My understanding is that ‘access’ is determined by various socioeconomic factors and different from availability.

Response: Thank you so much for the comments. Here, we tried to interlink the term access is the people’s capacity to purchase medicine safely according to their wish because around 80% of people in Bangladesh have to buy drugs from the shortly trained drug seller. 

5. Line 97: is not clear. What the author meant by density? Per facility?

Response: Thank you very much for the comment. Here, we used the ‘densely’ term from a reference that referred to the number of people (1045) living per square kilometer, not the per facility. We revised it accordingly. 

Page no.: 5, Line no.: 96-97

6. Line 99: “Following field study techniques of organizing complex research subjects and data, three topics were chosen as collection tools, namely 'places, people, and events.” Should be supported by evidence of where this technique has been used?

Response: We have corrected it. We mistakenly separated the sentences, but it was one sentence and had the evidence at the end of the sentence. 

Page no.: 5, Line no.: 102-103.

7. Line 186-187: “The opening hours of these drug stores were between 8.00 am and 10.am, however, they visited customers at home in an emergency.” This line is not clear. What the author wanted to mean here?

Response: Thank you very much for the comment. We reorganized the sentences. The first sentence is about the opening to closing hours. The second sentence was about the drug sellers visiting the client’s residence during the emergency call by the client. 

Page no.: 10, Line no.: 191-192.

8. Line 181: the author mentioned that “three different drug stores were selected from three different settings?” It is not clear how the author determined different settings in terms of what? and why the author have chosen three different settings?

Response: Thank you so much for the comment. We have revised it. The settings of the drug shops were selected based on the location of the drug store, sellers training and clients flow. 

Page no.: 10, Line no.: 183-187.

9. Line 195: author mentioned that “Customers’ drug purchasing behavior was influenced by demographic factors such as income, education, cultural beliefs and social network of the community people.” it is predictable that these socioeconomic factors influence customers’ drug purchasing behavior. However, what is new here? It would be useful to identify to what extent which factors affect the behaviour of the consumers? Otherwise, this statement remains vague, and dose not add anything to the findings. I assume this may not be possible to quantify this finding from only 30 interviews. I would recommend author not to focus on only the factors those affect the consumer drug purchasing behavior rather identify the key reasons for the identified behavior.

Response: Thank you so much for the comments. Yes, we all agreed with your suggestions and deleted the demographic factors. We revised following your comments. 

Page no.: 10-11, Line no.: 199-201.

10. Line 196: “Those living in wealthier areas mostly bought prescription and trade name drugs” I do not see any calculation of wealth quintile. How the author concluded this is not clear. A clear representation of the calculation of wealth quintile should be presented. Or on what criterion the author divided the respondents into different socioeconomic group should be mentioned clearly.

Response: Thank you so much for the comments. We are extremely sorry for the term. The term wealthier replaced into the apartment building living areas as several drug sellers mentioned that. 

Page no.: 11, Line no.: 201-202.

11. Line 207-208: please see comment 09.

Response: Thank you so much for the comment. We revised the sentences as ‘practices’ in place of patterns and ‘reasons’ in place of factors following the comment.

Page no.: 12, Line no.: 209, 212 

12. Line 225: “customer’s individual choices” I am not sure how the author determined the choices of consumer? Did the author wanted to mean their previous behaviour of drug purchasing?

Response: Thank you so much. We have checked our data and revised the term ‘wishes’ in place of choice. Thanks again. 

Page no.: 13, Line no.: 233.

13. Line 246: “In addition, the customer also seeks to buy a package of expensive drugs by instalment payments as well as by buying recommended drugs and those for a particular ailment.” This finding is interesting. However, the author ha not mentioned what is the understanding between the consumer and the drug seller regarding EMI? How the expensive drugs will be sold to consumers in EMI without any guarantee of repayment by the consumers?

Response: Thank you very for much for the comment. We revised and included about the understanding of medicine installment package according to our field data. Our data revealed that the client do not buy full package of medicine at once but rather buy on daily basis installment with cash payment. 

Page no. 14: Line no.: 256-259. 

14. Line 252: “around $3 @ 85 takas per $” The author should mention which dollar they referred to.

Response: We have added the USD in the revised version. Page no.: 14, Line no.: 263.

15. Line 278: “Findings revealed that some unregistered local drugs were dispensed and sold in the drug store.” I think it is difficult to conclude this finding from information from only three drug stores.

Response: Thank you very for the comment. We revised this following the field data that some of the drug sellers dispensed and sold some unregistered local medicines that are not listed (registered) in the drug administration.

Page no.:15, Line no.: 289-290. 

16. Line 406: “patient preference” the preference is a technical term, which should be assessed as revealed or stated. I would rather suggest the author to use behavior instead of preference.

Response: We revised this as ‘behavior’ instead of preference following your suggestion. Thank you so much.

Page no.: 21, Line no.: 419.

We sincerely thank you for the constructive comments and the opportunity to revise our manuscript. We sincerely hope that our responses are acceptable to you and the revised manuscript meets the PLOS ONE standard. If you have any issues with our answers or the revised manuscript, please let us know if I can further serve you. We will follow our guidance accordingly. Again, please allow us to express our gratitude for allowing us to revise our manuscript for consideration for publication in PLOS ONE. 

We uploaded the files namely cover letter, manuscript (clear version), Revised articles with changes highlighted, response to reviewers, and a request letter of permission for research data collection.

Best regards

Md. Shahgahan Miah, Ph.D.

Email: shahgahan-anp@sust.edu

Department of Society and Health

Faculty of Social Sciences and Humanities

Mahidol University, Thailand

---

## [Editor Report · Decision Letter 1]

16 Jan 2023

PONE-D-22-12723R1The Reality of Embedded Drug Purchasing Practices: Understanding the Sociocultural and Economic Aspects of the Use of Medicines in BangladeshPLOS ONE

Dear Dr. Md. Shahgahan Miah,

Thank you for submitting your manuscript to PLOS ONE. After careful consideration, we feel that it has merit but does not fully meet PLOS ONE’s publication criteria as it currently stands. Therefore, we invite you to submit a revised version of the manuscript that addresses the points raised during the review process.

We look forward to receiving your revised manuscript.

Kind regards,

Dr Haribondhu Sarma, PhD

Academic Editor

PLOS ONE

Journal Requirements:

Additional Editor Comments:

Apologies for taking the time to review your paper. Unfortunately, there is a limited number of reviewers who have the expertise and interest to review this paper. We have reviewers who initially agreed but later did not complete the review on time. I reviewed your paper and have following comments:

This is an interesting and important study that investigated the drug purchasing practices which underlie the socio-cultural and economic aspects of a Bangladeshi city. However, there are a few areas in the manuscripts that can be further improved considering my additional comments. I have following minor observations, please consider them while revising it:

1. In the first sentence of the abstract, please replace the word ‘standard’ with ‘common’. Please note that purchasing drugs with or without a prescription from retail drug shops is NOT a standard practice in Bangladesh, but it may be a common practice.

2. Please add a sentence under the results of the abstract with the number of IDIs, KIIs and observations you have conducted.

3. In your first revision, you did not highlight the texts in which you have made changes. Please do so in your next revision.

4. Line 77, please define what it means about ‘C’ grade pharmacists.

5. Line 190, please check again the time of the opening hours, whether it is 8.00 AM to 10.00 PM?

6. Please confirm whether the study has been approved by any local ethics committee.

---

## [Author Response · Author response to Decision Letter 1]

20 Feb 2023

Response to Academic Editor

Comment

Journal Requirements:

Response

Thank you so much for the comment and suggestions. In response to your observation, we conducted an additional check, and the reference has been updated. At this point, it is hopefully complete and correct according to the requirements of the journal. For your kind information, in our last revised manuscript, we included some new articles according to the reviewer's comments. That is why the reference number changed, and we apologize for not mentioning it in our last response and rebuttal letter. Once again, I would like to thank you for your comments and suggestions.

Additional Editor Comments:

Apologies for taking the time to review your paper. Unfortunately, there is a limited number of reviewers who have the expertise and interest to review this paper. We have reviewers who initially agreed but later did not complete the review on time. I reviewed your paper and have following comments:

This is an interesting and important study that investigated the drug purchasing practices which underlie the socio-cultural and economic aspects of a Bangladeshi city. However, there are a few areas in the manuscripts that can be further improved considering my additional comments. I have following minor observations, please consider them while revising it:

Response

Thank you so much for considering our manuscript for your list and for your appreciation. Yes, we tried to address all your comments and observations, which really improved our manuscript.

Comment: 1. 

In the first sentence of the abstract, please replace the word ‘standard’ with ‘common’. Please note that purchasing drugs with or without a prescription from retail drug shops is NOT a standard practice in Bangladesh, but it may be a common practice.

Response

Thank you so much for your valuable suggestion. Yes, we replaced it following your suggestion.

Page: 1 Line: 14

Comment:2.

Please add a sentence under the results of the abstract with the number of IDIs, KIIs and observations you have conducted.

Response: 

Thank you so much for the comment. We double checked and found that we included the number of IDIs and KIIs in the methods section of the abstract. Yes, we added a sentence about the observation hours that we conducted during our fieldwork. Thanks again for the comment.

Page: 1 Line: 22-23

Comment: 3. 

In your first revision, you did not highlight the texts in which you have made changes. Please do so in your next revision.

Response:

Thank you so much for the comment, and my apologies for the mistake. We followed your comments in our revised manuscript.

Comment:

Page: 4. Line 77, please define what it means about ‘C’ grade pharmacists.

Response: 

Thank you so much for the comment. Yes, we included a sentence that explained the types of pharmacists in Bangladesh.

Page: 4 Line: 80-83

Comment: 5. 

Line 190, please check again the time of the opening hours, whether it is 8.00 AM to 10.00 PM?

Response:

Yes, we double-checked and revised the opening hours. All three drug stores we investigated had regular hours of operation ranging from 8 a.m. to 10 p.m. However, they frequently open early and close late depending on the emergency needs of their clients. Thank you so much for the comment.

Page: 10 Line: 193-195

Comment:6.

Please confirm whether the study has been approved by any local ethics committee.

Response:

Thank you so much for your comment. When I submitted the data collection request letter approved by the faculty of graduate studies (FGS), Mahidol University addressed it to the civil surgeon (head of health services at the district level). He wanted to check my ethical permission paperwork and rebate my local ethical approval after reviewing the documents, as I had Mahidol University ethical (IRB) approval.

---

## [Decision Letter · Decision Letter 2]

6 Apr 2023

PONE-D-22-12723R2The Reality of Embedded Drug Purchasing Practices: Understanding the Sociocultural and Economic Aspects of the Use of Medicines in BangladeshPLOS ONE

Dear Dr. Md. Shahgahan Miah,

Thank you for submitting your manuscript to PLOS ONE. After careful consideration, we feel that it has merit but does not fully meet PLOS ONE’s publication criteria as it currently stands. Therefore, we invite you to submit a revised version of the manuscript that addresses the points raised during the review process.

ACADEMIC EDITOR: Minor Revision *Please provide a final paper with all revisions made and with an additional check on the level of plagiarism and compliance with the editorial Journal's guidelines.*

We look forward to receiving your revised manuscript.

Kind regards,

Vincenzo Basile, PhD

Academic Editor

PLOS ONE

Journal Requirements:

Reviewers' comments:

Reviewer's Responses to Questions

**Comments to the Author**

1. If the authors have adequately addressed your comments raised in a previous round of review and you feel that this manuscript is now acceptable for publication, you may indicate that here to bypass the “Comments to the Author” section, enter your conflict of interest statement in the “Confidential to Editor” section, and submit your "Accept" recommendation.

Reviewer #1: All comments have been addressed

Reviewer #2: (No Response)

2. Is the manuscript technically sound, and do the data support the conclusions?

Reviewer #1: Partly

Reviewer #2: Partly

3. Has the statistical analysis been performed appropriately and rigorously? 

Reviewer #1: I Don't Know

Reviewer #2: N/A

4. Have the authors made all data underlying the findings in their manuscript fully available?

Reviewer #1: Yes

Reviewer #2: No

5. Is the manuscript presented in an intelligible fashion and written in standard English?

Reviewer #1: Yes

Reviewer #2: No

6. Review Comments to the Author

Reviewer #1: Many thanks for addressing the earlier comments, I have some further comments and suggestions attached in the mansucript.

Reviewer #2: PONE-D-22-12723R2 review

The manuscript outlines a qualitative research study that explores drug seller and drug purchaser perspectives in Bangladesh. As I have stated, below, in my comments to the authors, literature on the topic from Bangladesh has not be explored and therefore there is overlap with previous findings. The authors need to revise their literature review and re-frame the introduction. This will allow the authors to focus on new findings rather than be repetitive of previous qualitative studies conducted in Bangladesh. I have added some papers for consideration but this is in no way exhaustive, just the ones with which I have some familiarity.

There are several comments on improving the English expression and adding more detail to improve manuscript clarity.

Because there are new findings that are worth highlighting, I think that the manuscript may be appropriate for publication after a significant revision.

Abstract

Line 22: are medical representatives pharmaceutical company employees? Please provide a more intuitive description to the international reader.

Line 23: nonverbal behaviour is a vague term. Add some specifics. The next sentence implies that a single participant was selected. Please revise this and indicate the total number.

Line 28: use the active voice. You are describing the results overall here. So, starting a sentence with 'results revealed' is not informative. In qualitative studies it is useful to include the type of data collection method that the results were drawn from. Do you mean 'We detected that xxxxxx from IDI thematic content analysis?'

Line 29: this sentence contradicts the first. You mean to describe the range of intentions of the customers. Revise the English expression to reflect this.

Line 31: it would be more useful to include at least one cultural belief. Using the term 'cultural belief' is vague and general.

Line 35: It is impossible to determine which were common versus rare behaviours. A semi-quantitative approach would be more meaningful. Some, most, many (with number of respondents) is often used in qualitative studies to indicate common vs. rare behaviour.

Line 37: the conclusion does not follow from the findings included in the results section. Be specific about what the findings can tell policy makers. This conclusion could have been written without conducting the study.

Introduction

Line 86: I disagree that there is little research on sociocultural and economic factors that contribute to economic aspects of drug purchasing. Quite a bit has been done on antibiotic purchasing alone, in Asia (including Bangladesh) and Africa, among others. A more detailed literature review is needed. This probably should have been done before developing IDI question guidelines.

Line 90: change 'gather' to 'gathered',

Methods

Line 109: it is known that different types of customers visit at different times of the day. Please provide some information on timing of preliminary observations.

Line 118: Since there were only three outlets, I am assuming that the authors mean 'one each of a seller with limited training, a RMP and a trainee sales assistant.

Line 122: this information should be in the results, not methods.

Line 129: please provide the number of each participant type.

Line 130: by primary healthcare providers with drug sellers, do you mean doctors that have a room in the drug shop??

Line 141: For an international audience, some context or photo of what comprises a drug shop in Bangladesh would be useful. For many in the Western world, they think this means a fully heated/air-conditioned 'pukka' premise with tiled floors etc.

Line 146: what questions examined economic situations of the participants? This is a finding that you have described. Was affordability a topic that came up without a prompt?

Line 148: what do you mean by local cultural drug purchasing?

Line 149: as indicated for the abstract, define what is meant by non-verbal interactions (note that this is in the manuscript both with and without a hyphen- ensure consistency throughout)

Line 154: what is meant by 'member checking'.

Ethical issues: it is uncommon for a study conducted in Bangladesh to not receive any local ethical approval. That form another country is often considered inadequate.

Results

Table 1:

Omit the first sentence. Start with the total number of participants then briefly describe their characteristics without repeating all of the data in the table.

some numbers are in bold and others not. I don't see a consistent pattern.

were there no data on where the key informants resided?

I think that a lay patient is just a patient.

age 45+/-9 seems to be in the wrong column

for education, it would be better if the year categories lined up for IDI and KII

Line 195: approximate or typical closing hours would be useful to include.

Line 203: It is difficult to determine whether behaviours described here and elsewhere in the manuscript were self-reported during IDIs or observed by researchers or both.

Line 205: The new sentence starting here should be part of a new paragraph. This information is about socioeconomic differences.

Table 2:

this is a description of the drug stores. Change the title to reflect this.

Since there are only three drug stores (sometimes called retail drug shops, drug shops- be consistent throughout the manuscript), it should be 'slum area' rather than 'slum areas'.

It's not clear where the data in this table were derived from. Drug shop staff or customers or both?

change common health problem to commonly reported health problem.

change 'selling products' to 'products sold'. Were these data from customers or drug shop staff or both.

Line 214 'Practices of' can be deleted.

Line 216: this reads as if there is one individual. Fix the English.

Line 218: Make the distinction between drugs for chronic conditions such as diabetes, hypertension etc and drugs for acute conditions e.g. infections. This is an important distinction.

Line 233: was this behaviour restricted to the slum dwellers or those visiting drug shop 2 only? It seems to be a common behaviour.

Line 250: include whether this was the case for all drug shops.

Line 283": this is not a good quotation. flucloxacillin is the name of the drug in fluclox'.

Line 293: do the authors mean the drug stores in the slum or the drug stores in the slum. Fix the English.

Line 296: remove the word 'of'.

Line 300: this is an interesting finding that drug sellers feel more comfortable to recommend unregistered drugs to those from lower SES groups. It's worth including in the discussion.

Line 304: omit the phrase 'from the drug store’. Also omit 'Data revealed that the'

Line 308: do the authors mean that these data came from drug shop observations? I have not seen any reference to findings from this data collection exercise up to this point in the manuscript.

line 320: circumcision is not just a procedure used by Muslims. It is also conducted by people from other religious groups and those without religious conviction. There's no need to describe this as a religious ritual.

Line 325: excellent quote.

Line 329: research data don't report, participants do. Fix this throughout. There are examples of ‘data showed’, results revealed etc. that need the expression revised.

Line 340: similarly use the active voice 'we found that customers often visited.......

Line 344: in the methods, the authors describe that observations are for detecting non-verbal interactions.

Line 346: there is a quotation mark missing. It should probably go after 'what should I do'.

Line 357: Competition to achieve a reputation for successful treatment: This business aspect of drug stores is well worth highlighting. Little attention is given to economic factors from the supply side. Often suggestions for antibiotic stewardship improvements, for example, assume that the supply side has bad motives. There’s nothing wrong in working to make a business thrive.

I did not find results on non-verbal interactions.

Discussion

The opening sentence is incorrect. See the following papers:

1. Lucas PJ, Uddin MR, Khisa N, Akter SMS, Unicomb L, Nahar P, Islam MA, Nizame FA, Rousham EK. Pathways to antibiotics in Bangladesh: A qualitative study investigating how and when households access medicine including antibiotics for humans or animals when they are ill. PLoS One. 2019 Nov 22;14(11):e0225270. https://journals.plos.org/plosone/article?id=10.1371/journal.pone.0225270

2. Nahar P, Unicomb L, Lucas PJ, Uddin MR, Islam MA, Nizame FA, Khisa N, Akter SMS, Rousham EK. What contributes to inappropriate antibiotic dispensing among qualified and unqualified healthcare providers in Bangladesh? A qualitative study. BMC Health Serv Res. 2020 Jul 15;20(1):656

3. Nizame FA, Shoaib DM, Rousham EK, Akter S, Islam MA, Khan AA, Rahman M, Unicomb L. Barriers and facilitators to adherence to national drug policies on antibiotic prescribing and dispensing in Bangladesh. J Pharm Policy Pract. 2021; 14(Suppl 1): 85. https://www.ncbi.nlm.nih.gov/pmc/articles/PMC8594093/

4. Matin MA, Khan WA, Karim MM, Ahmed S, John-Langba J, Sankoh OA, Gyapong M, Kinsman J, Wertheim H. What influences antibiotic sales in rural Bangladesh? A drug dispensers' perspective J Pharm Policy Pract. 2020 Jun 3;13:20.

Revise the first paragraph accordingly. Also, from these papers, determine what is new and novel about the data presented in the current manuscript. The first paragraph states that ‘new aspects’ were detected, which suggests that the authors are familiar with other work on this topic. This may possibly be from other countries. Nevertheless, it somewhat contradicts the opening sentence.

Line 379: this sentence is vague. Point to the results that support this sentence, which should probably be broken into two or more parts.

Line 387: this sentence is also not true, since there are published findings from Bangladesh and elsewhere that have reported this.

Line 399: I think this is a citation which is a superscript rather than in parentheses.

Line 403: this is new data that was not included in the results section. Move to results.

Line 434: purposive sampling is not a limitation but a standard qualitative technique.

Conclusion: see the comment for the conclusion section in the abstract.

7. PLOS authors have the option to publish the peer review history of their article (what does this mean?). If published, this will include your full peer review and any attached files.

Reviewer #1: No

Reviewer #2: No

---

## [Author Response · Author response to Decision Letter 2]

23 May 2023

Reviewers' comments:

Reviewer's Responses to Questions

Comments to the Author

1. If the authors have adequately addressed your comments raised in a previous round of review and you feel that this manuscript is now acceptable for publication, you may indicate that here to bypass the “Comments to the Author” section, enter your conflict of interest statement in the “Confidential to Editor” section, and submit your "Accept" recommendation.

Reviewer #1: All comments have been addressed

Reviewer #2: (No Response)

2. Is the manuscript technically sound, and do the data support the conclusions?

Reviewer #1: Partly

Reviewer #2: Partly

3. Has the statistical analysis been performed appropriately and rigorously?

Reviewer #1: I Don't Know

Reviewer #2: N/A

4. Have the authors made all data underlying the findings in their manuscript fully available?

Reviewer #1: Yes

Reviewer #2: No

5. Is the manuscript presented in an intelligible fashion and written in standard English?

Reviewer #1: Yes

Reviewer #2: No

6. Review Comments to the Author

Reviewer #2: 

Reviewer Comments and response:     

Reviewer 1

No. Comments Page and line number in the revised manuscript

1. Comment: 

Many thanks for addressing the earlier comments, I have some further comments and suggestions attached in the manuscript.

Response: 

Thank you so much for your valuable feedback to our manuscript. We tried our best to address all the comments and suggestions you made. 

N/A

2. Comment: 

PONE-D-22-12723R2 review

The manuscript outlines a qualitative research study that explores drug seller and drug purchaser perspectives in Bangladesh. As I have stated, below, in my comments to the authors, literature on the topic from Bangladesh has not be explored and therefore there is overlap with previous findings. The authors need to revise their literature review and re-frame the introduction. This will allow the authors to focus on new findings rather than be repetitive of previous qualitative studies conducted in Bangladesh. I have added some papers for consideration but this is in no way exhaustive, just the ones with which I have some familiarity.

There are several comments on improving the English expression and adding more detail to improve manuscript clarity.

Because there are new findings that are worth highlighting, I think that the manuscript may be appropriate for publication after a significant revision.

Response: 

Thank you very much for your comments and suggestions. We truly agree that the manuscript needs significant revisions with clear point and focus. We carefully revised it more consistently and respond to all of the comments and suggestions that you made. We really hope the revised manuscript addresses your concern.

We additionally revised some issues, including the affiliation of the first author. As a corresponding author, we did this along with some grammatical edits. 

N/A

Abstract

1. Comment: 

Line 22: Are medical representatives pharmaceutical company employees? Please provide a more intuitive description to the international reader.

Response: 

Thanks for your helpful suggestion. We revised this as pharmaceutical company representative in place of medical representative. 

 Page no. 1

 Line no. 22

2 Comment: 

Line 23: 

a. nonverbal behaviour is a vague term. Add some specifics.

 b. The next sentence implies that a single participant was selected. Please revise this and indicate the total number.

Response: 

We are really sorry for making you confused by the unclear sentence. The sentence is modified to make it more evident and to ensure clarity. 

a. We observed the conversations and interactions for medicine purchases between the drug seller and buyer. 

b. In particular, ‘participant’ here was mistakenly remained singular, so, we now corrected it to ‘participants’ and included the total number ‘A total of 40’. Additionally, we revised the last word analyzed thematically.

We hope that you find this revision clear. 

 Page no. 1 

 Line no. 23

 Page no. 1 

 Line no. 23-24, 25

3 Comment: 

Line 28: use the active voice. You are describing the results overall here. So, starting a sentence with 'results revealed' is not informative. In qualitative studies it is useful to include the type of data collection method that the results were drawn from. Do you mean 'We detected that xxxxxx from IDI thematic content analysis?'

Response: 

Yes, we have revised the sentence which included the ‘we found through thematic analysis following your suggestions. Thank you so much for the comments and suggestions. 

 Page no. 2

 Line no. 28

4 Comment: 

Line 29: this sentence contradicts the first. You mean to describe the range of intentions of the customers. Revise the English expression to reflect this.

Response: 

Yes, the revised sentence included ‘among the 30 participants, most individuals come without…”. Thank you so much for the suggestions. 

 Page no. 2 

 Line no. 29

5 Comment: 

Line 31: it would be more useful to include at least one cultural belief. Using the term 'cultural belief' is vague and general.

Response: 

We double checked our data and include the cultural beliefs of buying medicines in full or partial course of doses, with or without prescription following your suggestion.

For better clarity, we also revised the sentence ..experiences of medications shape the… 

 Page no. 2

 Line no. 31-32

 Line: 33

6 Comment: 

Line 35: It is impossible to determine which were common versus rare behaviours. A semi-quantitative approach would be more meaningful. Some, most, many (with number of respondents) is often used in qualitative studies to indicate common vs. rare behaviour.

Response: 

Thank you so much for your comment. We feel that putting the number of responses will clearly state the reflection of data. In this section, we revised the sentences following your suggestions and field data. We included few customers (n=7) sought drugs… a. many of the clients (n = 13) bought drugs through installment payments and with loans. 

 Page no. 2

 Line no. 34,36

 Comment: 

Line 37: the conclusion does not follow from the findings included in the results section. Be specific about what the findings can tell policy makers. This conclusion could have been written without conducting the study.

Response: 

Yes, we revised the conclusion in the abstract following your comments and suggestions. We tried to reorganize the conclusion in line with the study aim. 

 Page no. 2

 Line no. 39-43

Introduction

 Comment: 

Line 86: I disagree that there is little research on sociocultural and economic factors that contribute to economic aspects of drug purchasing. Quite a bit has been done on antibiotic purchasing alone, in Asia (including Bangladesh) and Africa, among others. A more detailed literature review is needed. This probably should have been done before developing IDI question guidelines.

Response: 

Thank you for your comment. In the revised version, we deleted the word little research and revised the sentence following the suggestions. 

We reviewed the literature you suggested, thanks for your suggestion. From this literature review, we have added relevant information to contextualize the research. 

 Page no. 5

 Line no. 95

Page no. 4, 5

Line no. 71-73, 93-95

 Comment: 

Line 90: change 'gather' to 'gathered',

Response: 

Thank you so much for the comment and suggestion. We have corrected it. 

 Page no. 5

 Line no. 99

Methods

 Comment: 

Line 109: it is known that different types of customers visit at different times of the day. Please provide some information on timing of preliminary observations.

Response: 

Thank you for your suggestion. We double checked our data and included drug purchasing timing information as the reviewer suggested.

Page no. 6

 Line no. 122-125

 Comment: 

Line 118: Since there were only three outlets, I am assuming that the authors mean 'one each of a seller with limited training, a RMP and a trainee sales assistant.

Response:

Thank you for your understanding and feedback. We have revised the sentence included one each of a seller with ‘C’ (3 months training) grade pharmacist, rural medical practitioner (RMP) and trainee sales assistant following the comment. 

 Page no. 6 

 Line no. 132- 136

 Comment: 

Line 122: this information should be in the results, not methods.

Response: 

Thank you so much for the comment and suggestion. We removed this from the methods following the suggestions as this information already in the result section. 

 Page no. 12

 Line no. 222-225

 Comment: 

Line 129: please provide the number of each participant type.

Response: 

Thank you for your suggestions. Yes, we provided the number of each participant type. 

 Page no. 7 

 Line no. 143- 145

 Comment: 

Line 130: by primary healthcare providers with drug sellers, do you mean doctors that have a room in the drug shop??

Response: 

Thank you for the query. Apology for the mistake and have corrected primary healthcare providers with drug sellers as drug sellers. We also corrected lay patients as patient, very experienced sales assistant as experienced sales assistants in the text and table as well. 

 Page no. 7, 9 

 Line no. 143-145

 Comment: 

Line 141: For an international audience, some context or photo of what comprises a drug shop in Bangladesh would be useful. For many in the Western world, they think this means a fully heated/air-conditioned 'pukka' premise with tiled floors etc.

Response: 

Thank you so much for your feedback. We included four photos of a drug store. 

We double-checked and included information about the context of drug stores in the revised version. 

 Page no. 8, 11

 Line no. 151-152, 

 Line no. 209-211

 Comment: 

Line 146: what questions examined economic situations of the participants? This is a finding that you have described. Was affordability a topic that came up without a prompt?

Response: 

Yes, we revised the sentence following the comments as we had questions about cost (ID guideline) of drug purchasing experiences to understand the economic situations of the participants. Thank you for the comment and suggestion. 

 Page no. 8

 Line no. 161

 Comment: 

Line 148: what do you mean by local cultural drug purchasing?

Response: 

Thank you so much for the query. We explained about the local cultural beliefs of medicine purchasing practices of buying full or partial course of drug dose with or without prescription. 

 Page no. 8 

 Line no. 163-164

 Comment: 

Line 149: as indicated for the abstract, define what is meant by non-verbal interactions (note that this is in the manuscript both with and without a hyphen- ensure consistency throughout)

Response: 

Apology for the mistakes and we deleted non-verbal term. Thank you so much for the comment and suggestions 

 Page no. 8

 Line no. 165

 Comment: 

Line 154: what is meant by 'member checking'.

Ethical issues: it is uncommon for a study conducted in Bangladesh to not receive any local ethical approval. That form another country is often considered inadequate.

Response: 

Thank you so much for the query. We mean the member checking about participant validation to ensure the data consistency with participants. 

In our past reviewer comments, we responded that we showed our ethical approval letter and the data collection request letter approved by the faculty of graduate studies (FGS), Mahidol University, addressed to the civil surgeon (head of health services at the district level, Bangladesh). The civil surgeon rebates our local ethical approval after reviewing the documents, as I had Mahidol University ethical (IRB) approval. 

 Page no. 9 

Line no. 170-171

 N/A

Results

 Comment: 

Table 1:

• Omit the first sentence. Start with the total number of participants then briefly describe their characteristics without repeating all of the data in the table.

• some numbers are in bold and others not. I don't see a consistent pattern.

• were there no data on where the key informants resided?

• I think that a lay patient is just a patient.

• age 45+/-9 seems to be in the wrong column for education, it would be better if the year categories lined up for IDI and KII

Response: 

• Yes, we have deleted the first sentence and started with to total number of participants as you suggested. 

• We non-bolded all the number in the table.

• We double checked and found the data about key informant residence where 8 were resided in suburb and 2 were in slum. 

• Yes, we revised the lay patient as only patient following your suggestion.

• We revised the table following your suggestions. Thanks again for your comments and suggestions.

Thank you so much for the comment and suggestion. 

 Page no. 10

 Line no. 189

 Line no. 200

 Comment: 

Line 195: approximate or typical closing hours would be useful to include.

Response: 

We checked our data and revised the approximate closing hours was 11p.m. Thank you so much for the suggestion. 

 Page no. 12

 Line no. 214

 Comment: 

Line 203: It is difficult to determine whether behaviours described here and elsewhere in the manuscript were self-reported during IDIs or observed by researchers or both.

Response: 

Thank you so much for the comments. Yes, we agreed and revised the sentence. We included the way customer purchase medicine is influenced by the cultural beliefs about medicine practices… 

 Page no. 12 

 Line no. 221

 Comment: 

• Line 205: The new sentence starting here should be part of a new paragraph. This information is about socioeconomic differences.

Table 2:

• this is a description of the drug stores. Change the title to reflect this.

Since there are only three drug stores (sometimes called retail drug shops, drug shops- be consistent throughout the manuscript), it should be 'slum area' rather than 'slum areas'.

• It's not clear where the data in this table were derived from. Drug shop staff or customers or both?

• change common health problem to commonly reported health problem. change 'selling products' to 'products sold'. Were these data from customers or drug shop staff or both.

Response: 

Thank you so much for the comments and suggestions. 

• Yes, we revise the paragraph following your suggestions. 

• Yes, we changed the title of the table 2 as demographics of retail drug stores: insights from drug shop staff and customers in accordance your comments. Apology for the inconsistencies. Yes, we revised as retail drug stores in throughout the manuscript.

• Yes, we have corrected the word slum areas into slum area

• Yes, we have corrected the word common health problem into commonly reported health problem, and selling products into products sold following the suggestions. We double checked with our data and included the retail drug shop staff and customers in the title of the table 2 as data derived from both drug shop staff and customers. 

 Page no. 13 

Line no. 226-227

 Line no. 233

 Comment: 

Line 214 'Practices of' can be deleted.

Response: 

Thank you for the comments. We revised the theme namely drug purchasing practices as different staff and customers involved in the process of buying and selling of medicine. 

 Page no. 14 

 Line no. 238

 Comment: 

Line 216: this reads as if there is one individual. Fix the English.

Response: 

Apology for the mistake. We revised this as apartment building residents in place of the individual living in the apartment building. Thank you so much for the comment. 

 Page no. 14 

 Line no. 240

 Comment: 

Line 218: Make the distinction between drugs for chronic conditions such as diabetes, hypertension etc and drugs for acute conditions e.g. infections. This is an important distinction.

Response: 

Thank you so much for your important suggestions. We checked our data and really sorry that we have no data about disease based medicine purchases. We will consider this in our further study. Thanks again. 

 N/A

 Comment: 

Line 233: was this behaviour restricted to the slum dwellers or those visiting drug shop 2 only? It seems to be a common behaviour.

Response: 

This behaviour reported among the low-income and illiterate customers in the drug shops 2 only. We included this in our revised version. 

 Page no. 15

 Line no. 260

 Comment: 

Line 250: include whether this was the case for all drug shops.

Response: 

We double checked our data and found that the case was for the drug shop 2 only and slum people considered this. We revised this according. 

 Page no 16 

Line no. 272

 Comment: 

Line 283": this is not a good quotation. flucloxacillin is the name of the drug in fluclox'.

Response: 

Thank you so much for the comments. This quotation reflects the effort that the drug seller put into convincing the customers to dispense their preferred brand of medicine if they asked for medicine with a brand name.

Yes, the medicine was the same, but different products were from different drug firms. We have corrected the flucloxacillin to G-flucloxacillin (produced by Gonoshasthya Pharma) after checking the data. Flucox, produced by the ACI Pharma, In Bangladesh, medicine is prescribed, sold, and purchased based on the drug’s brand name instead of its generic name. Therefore, drug sellers often dispensed more profitable and low-profile brand drugs, saying they were the same medicine. As only the top 20 drug firms maintain good manufacturing practices (GMP) in Bangladesh, there might be chances of selling and buying substandard medicines. Thanks for the comment. 

 Page no. 18

Line no. 309-310

 Comment: 

Line 293: do the authors mean the drug stores in the slum or the drug stores in the slum. Fix the English.

Response: 

Thank you so much for the comments and suggestions. We included drug stores in the slum and revised the writing errors, changed ‘sole’ into ‘sold’ and ‘from’ into ‘with’. 

Page no. 18 Line no. 317- 318

 Comment: 

Line 296: remove the word 'of'.

Response: 

Yes, we deleted the word of following your suggestions. Thank you for the suggestion. 

 Page no. 18

 Line no. 323

 Comment: 

Line 300: this is an interesting finding that drug sellers feel more comfortable to recommend unregistered drugs to those from lower SES groups. It's worth including in the discussion.

Response: 

Thank you so much for the comments. Yes, we included this in the discussion section following your comments on our revised manuscript. 

 Page no. 24

Line no. 453-455

 Comment: 

Line 304: omit the phrase 'from the drug store’. Also omit 'Data revealed that the'

Response: 

Yes, we omitted both the phrases following your suggestions 

 Page no. 17

 Line no. 326

 Comment: 

Line 308: do the authors mean that these data came from drug shop observations? I have not seen any reference to findings from this data collection exercise up to this point in the manuscript.

Response: 

Thank you so much for the comments. Yes, these data came from the observations. For your kind information, we used other data from observations. 

 Page no. 17 , 20

 Line no. 307 

Page no. 165, 257, 282, 335, 371

 Comment: 

line 320: circumcision is not just a procedure used by Muslims. It is also conducted by people from other religious groups and those without religious conviction. There's no need to describe this as a religious ritual.

Response: 

We deleted muslim ritual following your suggestions. Thank you so much for the comments and suggestions. 

 Page no. 19

 Line no. 347

 Comment: 

Line 325: excellent quote.

Response: 

Thank you so much for appreciation. 

 N/A

 Comment: 

Line 329: research data don't report, participants do. Fix this throughout. There are examples of ‘data showed’, results revealed etc. that need the expression revised.

Response: 

Apology for the mistake. We revised this as our participants also reported. We tried to fix this throughout the manuscript. Thank you so much for the comments and suggestions. 

 Page no. 13, 15, 18, 19, 

Line no. 236, 254, 278, 348, 379

 Comment: 

Line 340: similarly use the active voice 'we found that customers often visited.......

Response: 

Revised this following your suggestions. Thank you so much. 

Page no. 20 Line no. 367

 Comment: 

Line 344: in the methods, the authors describe that observations are for detecting non-verbal interactions.

Response: 

Acknowledging our mistake about using the word non-verbal. We omitted these from the entire manuscript. Thank you so much for the comments. 

 N/A

 Comment: 

Line 346: there is a quotation mark missing. It should probably go after 'what should I do'.

Response: 

Thank you so much for the comment. Yes, we included this in our revised manuscript. 

 Page no. 20

 Line no. 373

 Comment: 

Line 357: Competition to achieve a reputation for successful treatment: This business aspect of drug stores is well worth highlighting. Little attention is given to economic factors from the supply side. Often suggestions for antibiotic stewardship improvements, for example, assume that the supply side has bad motives. There’s nothing wrong in working to make a business thrive.

Response: 

Yes, we agree with you that there is nothing wrong in working to make a business thrive but the drug seller explained antibiotic use may harm health. We checked with our data and included findings from the drug sellers that this competition might increase the over use of antibiotic. 

 Page no. 21

 Line no. 390

 Comment: 

I did not find results on non-verbal interactions.

Response: 

We deleted the term non-verbal from the revised manuscript. N/A

Discussion

 Comment: 

The opening sentence is incorrect. See the following papers:

Revise the first paragraph accordingly. Also, from these papers, determine what is new and novel about the data presented in the current manuscript. The first paragraph states that ‘new aspects’ were detected, which suggests that the authors are familiar with other work on this topic. This may possibly be from other countries. Nevertheless, it somewhat contradicts the opening sentence.

Response: 

We deleted the first sentence. We tried to follow the suggested references style for revising our discussion part. 

We revised the first paragraph according to your comments and suggestions. 

 Page no. 22-21 

Line no. 404-419 

 Comment: 

Line 379: this sentence is vague. Point to the results that support this sentence, which should probably be broken into two or more parts.

Response: 

We have deleted this sentence, we mainly focused on the practices of medicine sells and purchases. We tried to break down the sociocultural and economic issues of medicine practices in our revised manuscript. Thank you so much for the comment. 

 Page no. 22-23 

Line no. 404-411

 Comment: 

Line 387: this sentence is also not true, since there are published findings from Bangladesh and elsewhere that have reported this.

Response: 

We deleted the words new aspects. We also revised this paragraph following the comments. We found compared our findings with suggested references. 

 Page no. 22-23 

Line no. 416-424

 Comment: 

Line 399: I think this is a citation which is a superscript rather than in parentheses.

Response: 

Apology for the mistake. We revised this. For your kind information, the list of references reordered as we included new references following your suggestions. 

 Page no. 23

 Line no. 429

 Comment: 

Line 403: this is new data that was not included in the results section. Move to results.

Response: 

Thank you so much for the comment. Yes, this data already included in the results section. 

 Page no. 20 

Line no. 356-363

 Comment: 

Line 434: purposive sampling is not a limitation but a standard qualitative technique.

Response: 

Yes, we agreed with you and apologize for the mistake. We deleted this and revised this limitation of the study section. 

 Page no. 25 

 Line no. 467-469

 Comment: 

Conclusion: see the comment for the conclusion section in the abstract.

Response: 

Thank you so much for the comments and suggestions. We revised the conclusion section following the comments. Thanks again. 

 Page no. 23-24 

 Line no. 470-483 

 Additional response

For your kind information, we edited few issues in the discussion section-

• We have edited some grammatical issues in the discussion section.

 Reviewer 2 

 Abstract 

 Comment:

Line 39 ‘drug resistance’ not sure what the author meant here

Response:

Thank you for the comment. We revised this as reduce the effectiveness of medication instead of drug resistance for more clarity. 

 Page no. 2 

 Line no. 40-41

 Comment: Line 66 This line is not clear. Did the author want to mean "used to visit”?

Response: 

Thank you so much for the comment. Yes, we try to mean ‘used to visit’ and revised accordingly. 

 Page no. 3 

 Line no. 69

 Comment: Line 67 patients where?

Response:

It was, Patients in Bangladesh. Revised accordingly 

 Page no. 4 

 Line no. 73-74 

 Comment: Line 76 Can the author add any reference?

Response: 

Thank you so much for the comments. We revised the number of 1,06,919 and cited from the reference [20]. Apology for the mistake. Thanks for understanding. 

 Page no. 4

 Line no. 83

 Comment: 

Line 87 This is the first time, author introduced two new terms. It would be great if author can define what actually meant in this paper by sociocultural and economic aspects. Is there any evidence in research that these two factors influence drug purchasing behavior? Can author include any conceptual framework or refer that how these factors may influence individual behavior while purchasing drugs? What is the hypothesis and how it is grounded on theory?

Response: 

Thank you so much for your comments and suggestions. We used these two new terms in order to understand the medicine purchasing practices and how these are connected to sociocultural and economic aspects. What are the realities of expectation, negotiation, and decision-making, and how are medicines being dispensed and acquired between drug sellers and customers? The sociocultural and economic aspects of the hypothesis provide answers to these queries.

Sorry about the word of using factor. We included the word aspects instead of factor. As of our best knowledge, we didn’t find any evidence about these two factors influence drug purchasing behavior.

Yes, we included a conceptual framework that reflect the drug dispense and purchasing practices interlinked with the sociocultural and economic aspects.

Sorry, we did not used any theory in this research.

Thank again for the comments and suggestions. This really helps to make it more evident. 

 Page no. 5

 Line no. 95-96

 Page no. 12

 Line no. 224

 Comment: 

In line 87 What are the evidences that irrational drug buying exists?

Response: 

The past research citation no. [16] identified the unnecessary buying and selling of drugs. We replaced the word unnecessary in place of irrational. 

 Page no. 5

 Line no. 96 

 Comment: 

Line 95 The author did not mention anything about sociocultural and economic aspects rather than general description of qualitative analysis in the methodology. It may be mentioned how the author assessed these two aspects which are the prime focus of this manuscript?

Response: 

Thank you so much for the comment. We tried to mention briefly about the sociocultural and economic aspects in the methods section. As we adopted the ethnographic method, we tried to explore the describe in details practices about drug purchasing and sells. For your information, we did not assessed quantitative indicators except some demographic information of the participants. We analyzed the practices of medicine as behavioral part. Thanks again. 

 Page no. 5 

 Line no. 115-118

 Comment: 

Line 99 The author may clarify if there were any criterion for selecting study site?

Response: 

We already clarify about the study area that is densely populated and poorest healthcare indicator in Bangladesh. Thank you for the comment. 

 Page no. 5 

 Line no. 108-109 

 Comment: 

Line 117 This line is confusing.

Response: 

Apology for the mistake. We revised the sentence as participants were purposefully selected from three different retail drug stores where they worked as private merchants, primary healthcare professionals, or consumers. Thanks for the comment. 

 Page no. 6

 Line no. 132-134 

 Comment: 

Line 128-132 Categories of the respondents seemed do not match with the reported table.

Response: 

Yes, we revised table 1 accordingly. 

 N/A

 Comment: 

Table 1, What is the difference between clients and lay patients? Authors may consider two sides, supply (users) and demand (providers). Other classification is confusing.

Response: 

Thank you so much for comment and suggestion. We included this as clients and patients. Clients who bought medicine for his own or others and the patient who present symptoms and bought medicine. We did not categorize supply and demand sides. Thanks again for the suggestion, will consider for further study. 

 Page no. 10

 Line no. 200

 Comment: 

Line 185, This section seemed to be a mix of background, methods, and results in the result section, the author may reorganize it.

Response: 

Thank you so much for the comment. Yes, we agree with you. We tried to explain about the context through this mixed description. Thanks for the comment. 

 N/A

 Comment: 

Line 203 t is unclear how the authors concluded these findings? Where is the data? How many of the respondents mentioned these aspects? How the authors measured their cultural beliefs?

Response: 

Apology for the mistake. We deleted the customers drug purchasing behavior. We revised this as the way customer purchase medicine is influenced by…..

We did not measure their beliefs rather describe their cultural practices of buying medicine. 

 Page no. 12 

 Line no. 221

 Comment: 

Table 2 t may be but would be strong to say that a drug store being in an affluent area, their clients will be from higher income group. Did the authors analyzed the characteristics of the respondents who visited these drug stores? There might be several factors for a client to visit a drug store, one of them is drugs availability. Poor people may also visit to drug seller in the affluent area as they might sell drugs which are not available in other place or maybe there are other factors, easy to reach, give discounts. 

Response: 

Thank you so much for the comments and suggestions. As we did not measure the asset quantile, we just used the term higher income residential area considering the location of the drug store. We double-checked our data and found that the slum people used to visit their nearest drug store. 

 N/A

 Comment: 

Line 216 How did author obtained this information?

Response: 

We obtained this information from the demographic data in particular the residual status during interview Page no. 14

 Line no. 240

 Comment: 

Line 268 What if the drug is an antibiotic?

Response: 

Thank you so much for the comment. Similar experiences shared in case of antibiotic. We included the quotation in case of antibiotic. 

 Page no. 17 

 Line no. 288-292 

 Discussion 

 Comment:

Line 379 This is not well justified in this study how these linked with economic aspects. This should be clear in the method and grounded on any theory.

Response:

Yes, we deleted this and revised accordingly. Thank you so much for the comment. 

 Page no. 22 

 Line no. 404-408 

 Comment: 

Line 387, I am not sure if this is really a new aspect or regular practice of people with from low socioeconomic group.

Response: 

We revised this sentence and reorganized accordingly. Thank you so much for the comment and suggestion. 

 Page no. 22

 Line no. 415-419 

 Comment: 

In line 415 The author may also discuss the about the drug policy of Bangladesh and specifically where this study can contribute.

Response: 

Thank you so much for the comment. We just briefly include the monitoring and regulation of the over the counter drugs following the national drug policy. 

 Page no. 24 

 Line no. 446-447 

 Comment: 

In line 427 This is not clear how the patients will be aware from this findings? Should policymakers take initiatives to aware the patients?

Response: 

Yes, the policymakers could take the initiative of multilayer intervention to aware the patients. We found that drug sellers could play the vital role as health educator for the patients. 

 Page no. 23, 24

Line no. 435, 462-463

 Comment: 

In line 437 The author may mention theoretical aspects related to healthcare belief and how it is influenced such as using health belief model or revise related terminologies.

Response: 

We deleted the sentence and revised following the reviewers’ comments and suggestions. We did not adopt any theory in this study. Thank you so much for the comment. 

 Page no. 24 

 Line no. 467-469

 Comment: 

In line 461 This seemed not clear, what author exactly want to mean or suggest?

Response: 

Thank you so much for the comment. We tried to make it clear including the sentence as drug sells and purchases practices varied with urban settings of slum and suburb areas. Therefore, considering the local context of drug practices may implicate into the policy for ensuring safe and rational use of medicine. 

 Page no. 26

Line no. 492-493 

7. PLOS authors have the option to publish the peer review history of their article (what does this mean?). If published, this will include your full peer review and any attached files.

Do you want your identity to be public for this peer review? For information about this choice, including consent withdrawal, please see our Privacy Policy.

Reviewer #1: No

Reviewer #2: No

Again, thank you very much for all your meaningful comments and suggestions. We hope that all of our responses address your concerns.

Sincerely yours,

Dr. Md. Shahgahan Miah

Associate Professor

Department of Anthropology

Shahjalal Univeristy of Science and Technology

Sylhet-3114, Bangladesh

Email: shahgahan-anp@sust.edu

---

## [Editor Report · Decision Letter 3]

29 May 2023

The Reality of Embedded Drug Purchasing Practices: Understanding the Sociocultural and Economic Aspects of the Use of Medicines in Bangladesh

PONE-D-22-12723R3

Dear Dr. Md. Shahgahan Miah,

We’re pleased to inform you that your manuscript has been judged scientifically suitable for publication and will be formally accepted for publication once it meets all outstanding technical requirements.

Kind regards,

Vincenzo Basile, PhD

Academic Editor

PLOS ONE

Additional Comments:

*Please provide a final paper with all revisions made and I recommend an additional check on plagiarism and/or compliance with the Journal's guidelines.*

---

## [Editor Report · Acceptance letter]

9 Jun 2023

PONE-D-22-12723R3 

The Reality of Embedded Drug Purchasing Practices: Understanding the Sociocultural and Economic Aspects of the Use of Medicines in Bangladesh 

Dear Dr. Miah:

I'm pleased to inform you that your manuscript has been deemed suitable for publication in PLOS ONE. Congratulations! Your manuscript is now with our production department. 

Kind regards, 

on behalf of

Dr. Vincenzo Basile 

Academic Editor

PLOS ONE